# AUXILIARY-LOSS-FREE LOAD BALANCING STRATEGY FOR MIXTURE-OF-EXPERTS

## ABSTRACT

For Mixture-of-Experts (MoE) models, an unbalanced expert load will lead to routing collapse or increased computational overhead. Existing methods commonly employ an auxiliary loss to encourage load balance, but a large auxiliary loss will introduce non-negligible interference gradients into training and thus impair the model performance. In order to control load balance while not producing undesired gradients during training, we propose **Loss-Free Balancing**, a new load balancing strategy that operates without auxiliary losses. To be specific, before the top-K routing decision, Loss-Free Balancing will first apply an expert-wise bias to the routing scores of each expert. By dynamically updating the bias of each expert according to its recent load, Loss-Free Balancing can consistently maintain a balanced distribution of expert load. In addition, since Loss-Free Balancing does not produce any interference gradients, it also elevates the upper bound of model performance gained from MoE training. We validate the performance of Loss-Free Balancing on MoE models with up to 3B parameters trained on up to 200B tokens. Experimental results show that Loss-Free Balancing achieves both better performance and better load balance compared with traditional auxiliary-loss-controlled load balancing strategies.

## 1 INTRODUCTION

Mixture-of-Experts (MoE) architectures have emerged as a promising solution for managing computational costs when scaling up parameters in large language models (LLMs). Recent applications of MoE in Transformer-based models (Vaswani et al., 2017) have led to successful attempts at scaling language models to substantial sizes (DeepSeek-AI et al., 2024a;b; Dai et al., 2024; Fedus et al., 2022; Lepikhin et al., 2021), resulting in remarkable performance improvements. However, training MoE models always face the circumstance of load imbalance, which may result in routing collapse (Shazeer et al., 2017) or increased computational overhead (Fedus et al., 2022; Lepikhin et al., 2021; Shazeer et al., 2017). In order to avoid imbalanced routing, existing methods (Fedus et al., 2022; Lepikhin et al., 2021) commonly use an auxiliary loss to encourage balanced expert load. Although the auxiliary loss can alleviate load imbalance during training, it also introduces undesired gradients that conflict with the language modeling objective. These interference gradients will impair the model performance, so existing MoE methods always need to consider the trade-off between load balance and model performance.

In this paper, we propose **Loss-Free Balancing**, an auxiliary-loss-free load balancing strategy, aiming at maintaining control over expert load balance while not introducing interference gradients. Loss-Free Balancing features an iterative process of token routing and bias updating. As illustrated in Figure 1, before the top-K routing decision of MoE, Loss-Free Balancing will first apply expert-wise biases to the original routing scores to produce biased gating scores, which determine the actual routing targets of each token during training. These expert-wise biases will keep updating according to the expert load observed on recent training tokens, where the biases of heavy-load experts will be depressed and those of lite-load experts will be elevated. Through this dynamic updating strategy,

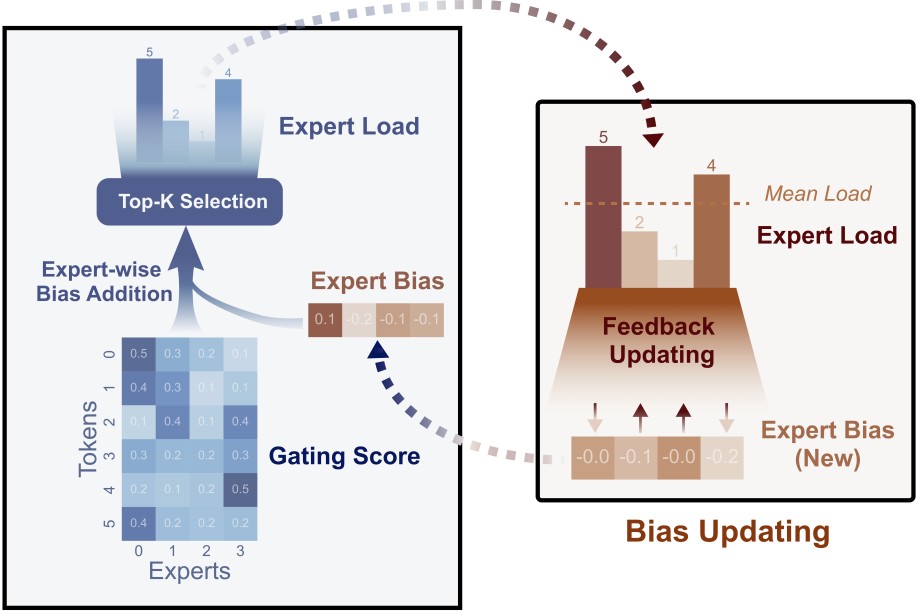

Figure 1: Loss-Free Balancing selects experts according to a "biased gating score" in each training step and updates this expert-wise bias after each training step.

Loss-Free Balancing ensures that the biased gating scores can consistently lead to balanced routing results. Compared with the auxiliary-loss-controlled load balancing strategies, Loss-Free Balancing does not introduce undesired gradients that disrupt the primary language modeling objective, so its training process is more noise-free and friendly.

In order to validate the performance of Loss-Free Balancing, we train MoE language models with 1B parameters on 100B tokens and 3B parameters on 200B tokens from scratch. Experimental results demonstrate that Loss-Free Balancing produces MoE models with better validation loss than traditional auxiliary-loss-controlled models. Meanwhile, keeping the performance advantage, Loss-Free Balancing also achieves a significantly better load balance at the global and batch levels, and is naturally compatible with expert parallelism, which is usually employed for training extremely large MoE models.

## 2 BACKGROUND

### 2.1 MIXTURE-OF-EXPERTS

Current dominant MoE architectures (Lepikhin et al., 2021; Fedus et al., 2022; Dai et al., 2024) replace the MLP layers in standard transformers with MoE layers. In an MoE layer, Top-K routing is employed to select the experts for each token. Let $\mathbf{u}_t$ denote the input of the $t$-th token to an $N$-expert MoE layer, the output $\mathbf{h}_t$ is computed as follows:

$$\mathbf{h}_t = \mathbf{u}_t + \sum_{i=1}^{N} g_{i,t} \operatorname{FFN}_i \left( \mathbf{u}_t \right),$$

$$g_{i,t} = \begin{cases} s_{i,t}, & s_{i,t} \in \operatorname{Topk}\left( \{ s_{j,t} \mid 1 \leq j \leq N \}, K \right), \\ 0, & \text{otherwise}, \end{cases} \tag{1}$$

$$s_{i,t} = G \left( \mathbf{u}_t^T \mathbf{e}_i \right),$$

where $G$ is a nonlinear gating function and $\mathbf{e}_i$ is the centroid of the $i$-th expert.

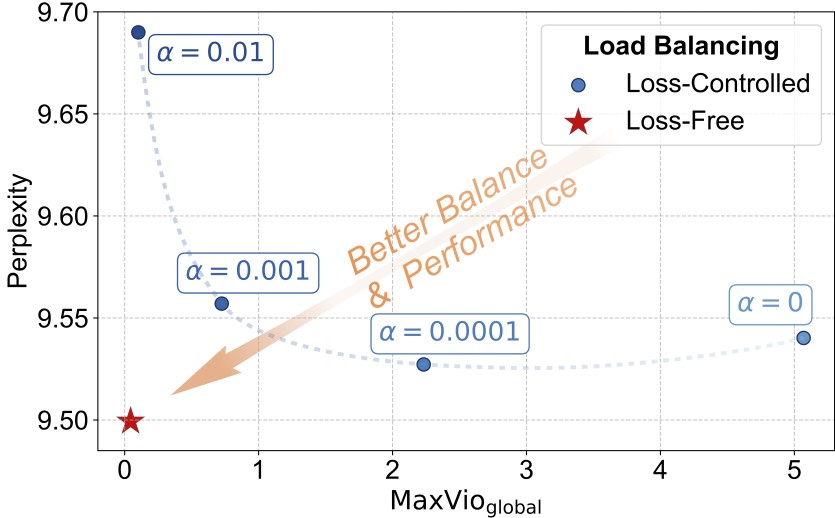

Figure 2: The dilemma between load balance and model performance for auxiliary-loss-controlled training. A small auxiliary loss coefficient $\alpha$ leads to poor load balance, while a large $\alpha$ impairs the model performance. In contrast, our Loss-Free Balancing method breaks this dilemma.

## 2.2 AUXILIARY LOSS FOR LOAD BALANCE

**Auxiliary Loss**   Uncontrolled routing strategies are likely to encounter load imbalance, which has two notable drawbacks. Firstly, there is a risk of routing collapse (Shazeer et al., 2017), where the model consistently selects only a few experts, hindering sufficient training of the other experts. Secondly, when experts are distributed across multiple devices, load imbalance can exacerbate computation bottlenecks. To address these issues, an auxiliary loss (Fedus et al., 2022; Lepikhin et al., 2021) is commonly employed to control load balance. For a sequence of length $T$, the auxiliary loss is defined as:

$$\mathcal{L}_{\text{Balance}} = \alpha \sum_{i=1}^{N} f_i P_i,$$

$$f_i = \frac{N}{KT} \sum_{t=1}^{T} \mathbb{1}(\text{ Token } t \text{ selects Expert } i), \tag{2}$$

$$P_i = \frac{1}{T} \sum_{t=1}^{T} s_{i,t},$$

where $N$ is the total number of experts, $K$ is the number of experts selected for each token, $s_{i,t}$ is the routing score of Expert $i$ for Token $t$, $f_i$ represents the fraction of tokens routed to Expert $i$, $P_i$ denotes the average gating scores of Expert $i$, and $\alpha$ is a hyper-parameter controlling the strength of the auxiliary loss.

**The Dilemma Between Load Balance and Model Performance**   The auxiliary loss mentioned above can encourage load balance, but it also interferes with language modeling training as an additional regularization term. The absence of an auxiliary loss or a small auxiliary loss coefficient $\alpha$ can lead to poor balance, while a large $\alpha$ can impair training, resulting in suboptimal performance. To illustrate this dilemma, we present the relationship between load balance and model performance in Figure 2. We vary $\alpha$ among 1e-2, 1e-3, 1e-4, and 0, and present the corresponding MaxVio$_{\text{global}}$, which measures the degree of load balance and its computation details are described in § 4.1. As shown in the figure, a small $\alpha$ causes routing collapse, affecting the model efficiency and potentially leading to some experts being insufficiently learned or exploited; while a large $\alpha$ keeps load balance

---

**Algorithm 1:** Adjusting the per-expert bias $b_i$ during training

**Input:** MoE model $\theta$, training batch iterator $B$, bias update rate $u$.

1. Initialize $b_i = 0$ for each expert;

**for** *a batch* $\{(\mathbf{x}_k, \mathbf{y}_k)\}_k$ *in* $B$ **do**

  2. Train MoE model $\theta$ on the batch data $\{(\mathbf{x}_k, \mathbf{y}_k)\}_k$, with gating scores calculated
   according to Eq. (3);

  3. Count the number of assigned tokens $c_i$ for each expert, and the average number $\overline{c_i}$;

  4. Calculate the load violation error $e_i = \overline{c_i} - c_i$;

  4. Update $b_i$ by $b_i = b_i + u * \mathrm{sign}(e_i)$;

**end**

**Output:** trained model $\theta$, corresponding bias $b_i$

---

under control but notably degrades the model performance. In order to break this dilemma, we propose **Loss-Free Balancing** as a solution, which directly controls the expert load balance, but does not introduce unexpected gradients other than the gradients from the language modeling loss.

## 3 AUXILIARY-LOSS-FREE LOAD BALANCING STRATEGY

For a better load-balancing alternative that does not directly interfere with the main gradients from the training objective, we propose **Loss-Free Balancing**, which directly adjusts the gating scores of each expert according to their balance condition. As illustrated in Figure 1, we add an expert-wise bias term $\{b_i\}_{i=1}^N$ to the gating scores $s_{i,t}$ of each expert, and use the biased scores to determine the top-K selection:

$$g_{i,t} = \begin{cases} s_{i,t}, & s_{i,t} + b_i \in \mathrm{Topk}\left(\{s_{j,t} + b_j \mid 1 \leq j \leq N\}, K\right), \\ 0, & \text{otherwise.} \end{cases} \tag{3}$$

Note that the expert bias term $b_i$ is only used to adjust the routing strategy by influencing the top-K selection. It is not added to the $g_{i,t}$ that weights the output of the selected experts when computing the final output of the MoE layer.

In order to derive proper biases, we adjust each bias $b_i$ iteratively according to the following principle: decreasing it when the corresponding expert has a relatively heavy load, and vice versa. To be specific, for each $b_i$, we keep monitoring its corresponding expert load on the previous batch. If an expert has a heavy load on the previous batch, we will reduce its bias. Otherwise, we will increase it. Algorithm 1 describes the details of our update algorithm for the expert-wise biases. It is worth noting that we update the biases based on the historical balance condition, since utilizing the load information of the current sequence will break the causal constraint of language modeling, leading to leakage of the information of future tokens. Through the dynamic adjustment for the biases, we can achieve good expert load balance, but not directly introduce noisy gradients into the model like the auxiliary-loss-controlled method does.

**Comparison with Other Load Balancing Methods.** In order to show the theoretical advantages of Loss-Free Balancing, we compare it with other two mainstream load balancing methods, i.e., the auxiliary-loss-controlled method (Lepikhin et al., 2021; Fedus et al., 2022) and the Expert Choice (EC) (Zhou et al., 2022) method. As described in § 2.2, the auxiliary-loss-controlled method faces the dilemma between load balance and model performance, and a perfect trade-off may not exist. As for the EC method, it will break the causal constraint of language modeling, since the target experts of each token are conditioned on the future tokens in the same sequence or batch. This will result in the leakage of information about future tokens, thus destroying the generalization of the model. Table 1 summarizes the properties of different load balancing methods.

Table 1: Comparison among different load balancing methods. The good property is displayed in green and the bad property in red.

| Load Balancing Methods | Balanced Expert Load | Interference Gradients | Future Token Leakage |
|---|---|---|---|
| Loss-Controlled (strong auxiliary loss) | balanced | strong | no leakage |
| Loss-Controlled (weak auxiliary loss) | imbalanced | weak | no leakage |
| Expert Choice | balanced | none | with leakage |
| Loss-Free (Ours) | balanced | none | no leakage |

## 4 EXPERIMENTS

### 4.1 EXPERIMENTAL SETUPS

**Model Architecture.**   We employ the DeepSeekMoE (Dai et al., 2024) architecture as the backbone since it outperforms conventional MoE architectures like GShard (Lepikhin et al., 2021) significantly. Compared with GShard (Lepikhin et al., 2021), it segments experts into finer granularity and isolates some experts as shared ones. Slightly different from DeepSeekMoE, in our main experiments, we choose sigmoid instead of softmax as the gating function $G$, since we find that the sigmoid baseline performs better than the softmax baseline. Even so, we still provide the experimental results and discussion for the softmax gate in Appendix C. Our experiments are based on two model sizes of 1B and 3B total parameters, and we tune the bias update rate under only the 1B scale. Experiments under the 3B scale directly inherit the best configuration for the 1B scale. Due to the page limit, we present more details about our architecture in Appendix A.

**Training Settings**   We use a multilingual training corpus created by DeepSeek-AI, sourced from a diverse range of textual materials including web text, mathematical material, coding scripts, and published literature. We employ the HuggingFace Tokenizer[1] to train a byte pair encoding (BPE) (Sennrich et al., 2016) tokenizer with a vocabulary size of 32K. In order to draw solid conclusions, we train the 1B model on 100B tokens and the 3B model on 200B tokens to ensure sufficient training. We apply the cosine learning rate scheduler (Loshchilov & Hutter, 2017) and multi-step learning rate scheduler (Dai et al., 2024) for the 1B and 3B models, respectively. Due to the page limit, we list more details about our training settings and hyper-parameters in Appendix B).

**Baseline.**   We compare our Loss-Free Balancing method with the conventional auxiliary-loss-controlled method. For the baseline, we set the auxiliary loss coefficient $\alpha$ to 0.001 to achieve a reasonable trade-off between model performance and load balance (see Figure 2). We do not take the EC method into comparison due to its issue of future token leakage, which we will discuss in depth in § 5.2.

**Metrics.**   We reserve a validation set from the training corpus to evaluate model performance and load balance. For model performance, we take perplexity as the metric. For load balance, we introduce a metric called maximal violation (**MaxVio**) to quantify the degree of load balance of an MoE layer:

$$\text{MaxVio} = \frac{\max_i \text{Load}_i - \overline{\text{Load}_i}}{\overline{\text{Load}_i}}, \tag{4}$$

where $\text{Load}_i$ represents the number of tokens assigned to the $i$-th expert, and $\overline{\text{Load}_i}$ denotes the expected expert load under perfect load balance.

---

[1]https://github.com/huggingface/tokenizers

Table 2: Loss-Free Balancing achieves lower perplexity and better load balance on both 1B and 3B models. A validation set is used to calculate these metrics (see details in Appendix B).

| Model Size | Load Balancing Methods | Validation Perplexity | MaxVio$_{\text{global}}$ |
|:---:|:---:|:---:|:---:|
| 1B | Loss-Controlled | 9.56 | 0.72 |
| | Loss-Free | **9.50** | **0.04** |
| 3B | Loss-Controlled | 7.97 | 0.52 |
| | Loss-Free | **7.92** | **0.04** |

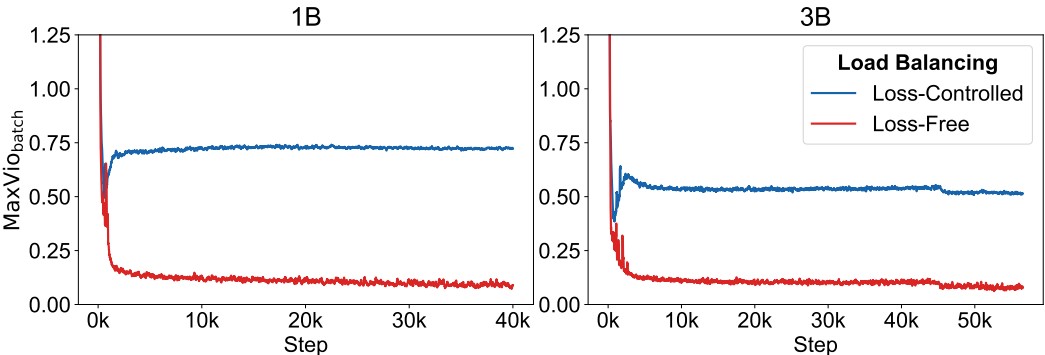

Figure 3: Loss-Free Balancing maintains a better load balance throughout most of the training time. Here, MaxVio$_{\text{batch}}$ is averaged over 100 neighboring steps for visibility purposes.

**MaxVio** has two variants: **MaxVio$_{\text{global}}$** and **MaxVio$_{\text{batch}}$**. For **MaxVio$_{\text{global}}$**, we count Load$_i$ on the whole validation set, so it reflects the degree of balanced expert utilization and efficiency upper bound when the batch size approaches the limitation. For **MaxVio$_{\text{batch}}$**, we count Load$_i$ on each training batch, so it is more related to the training efficiency. For simplicity, in the rest of this paper, we report the MaxVio averaged across all layers as a load balance measurement of the whole model.

### 4.2 MAIN RESULTS

Table 2 shows the validation perplexity and MaxVio$_{\text{global}}$ for the 1B and 3B MoE models trained with auxiliary loss or our auxiliary-loss-free load balancing strategy. As shown in the table, compared with the auxiliary-loss-controlled method, our Loss-Free Balancing achieves better perplexity and much better global load balance for both 1B and 3B models. In addition, to present the load balance condition during training, we provide a load balancing curve depicting MaxVio$_{\text{batch}}$ over training steps in Figure 3, which demonstrates the persistent advantage of Loss-Free Balancing on load balance. In summary, our Loss-Free Balancing method avoids interfering gradients during training and effectively controls the load balance, breaking the dilemma between load balance and model performance in MoE training.

### 4.3 EMPIRICAL STUDIES ON BIAS UPDATE ALGORITHM

We conduct empirical studies on the update rate and variants of the bias update algorithm to validate the optimal configuration used in our main experiments.

**Update rate.** The update rate $u$ in Algorithm 1 controls the speed at which the expert bias $\{b_i\}_{i=1}^N$ converges to the "suitable bias". Figure 4 illustrates that an overly low update rate $u = 0.0001$ may lead to slow convergence, while an unnecessarily high update rate $u = 0.01$ can cause undesirable fluctuations of the expert bias $b_i$ during the later stage of training, deteriorating load balance in this

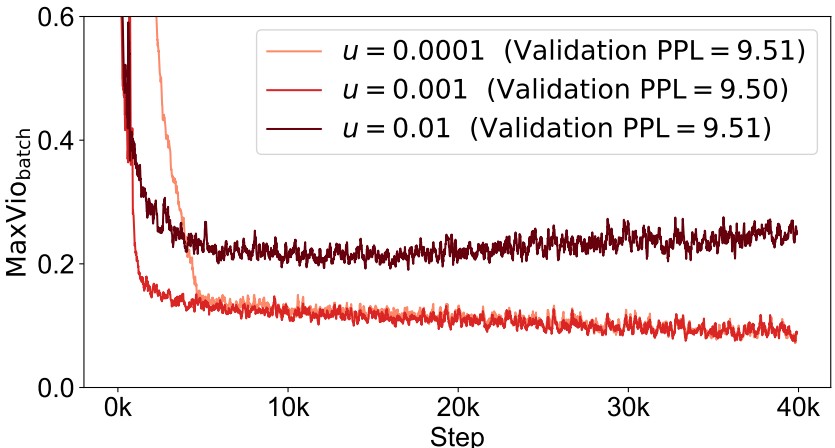

Figure 4: The impact of update rate on training load balance. A low update rate shows poor load balance in the early stage of training, while a high update rate deteriorates load balance in the later stage. Validation PPL denotes the validation perplexity.

Table 3: The variant $b_i = b_i + u * e_i$ slightly improves load balance but does not show improvement in model performance.

| Method | Perplexity | MaxVio$_{\text{global}}$ |
|---|---|---|
| $b_i = b_i + u * \text{sign}(e_i)$, $u = 0.001$ | **9.50** | 0.044 |
| $b_i = b_i + u * e_i$, $u = 0.01$ | 9.53 | **0.028** |
| $b_i = b_i + u * e_i$, $u = 0.001$ | 9.51 | 0.036 |
| $b_i = b_i + u * e_i$, $u = 0.0001$ | 9.51 | 0.040 |

stage. Both situations can impair performance. An appropriate choice is $u = 0.001$, which shows good training balance and validation perplexity.

**Update rule.** We investigate a different update rule of the expert-wise biases. To be specific, we attempt to change the update rule of $b_i = b_i + u * \text{sign}(e_i)$ to $b_i = b_i + u * e_i$, which encourages the bias of experts with high violation errors to change faster. Although this variant slightly improves load balance, it does not lead to better performance, as shown in Table 3. Therefore, we maintain the sign version.

**Multiplicative bias.** In addition to adding the expert-wise biases to the gating scores, using multiplicative biases is also a potential variant:

$$g_{i,t} = \begin{cases} s_{i,t}, & s_{i,t} * b_i \in \text{Topk}\left(\{s_{j,t} * b_j \mid 1 \le j \le N\}, K\right), \\ 0, & \text{otherwise}, \end{cases} \quad (5)$$

These $\{b_i\}_{i=1}^{N}$ can be updated using a similar procedure to Algorithm 1, except that they should be initialized as 1 instead of 0. Table 4 shows that using multiplicative biases results in slightly worse model performance compared to using additive biases, without significant improvements in load balance. Based on these findings, we conclude that additive biases are a more suitable choice for our method.

Table 4: Multiplicative bias shows similar load balance but slightly worse performance compared to additive bias.

| Method | Perplexity | MaxVio$_{global}$ |
|---|---|---|
| Addative Bias, $u = 0.001$ | **9.50** | 0.044 |
| Multiplicative Bias, $u = 0.01$ | 9.52 | 0.041 |
| Multiplicative Bias, $u = 0.001$ | 9.52 | **0.036** |
| Multiplicative Bias, $u = 0.0001$ | 9.54 | 0.048 |

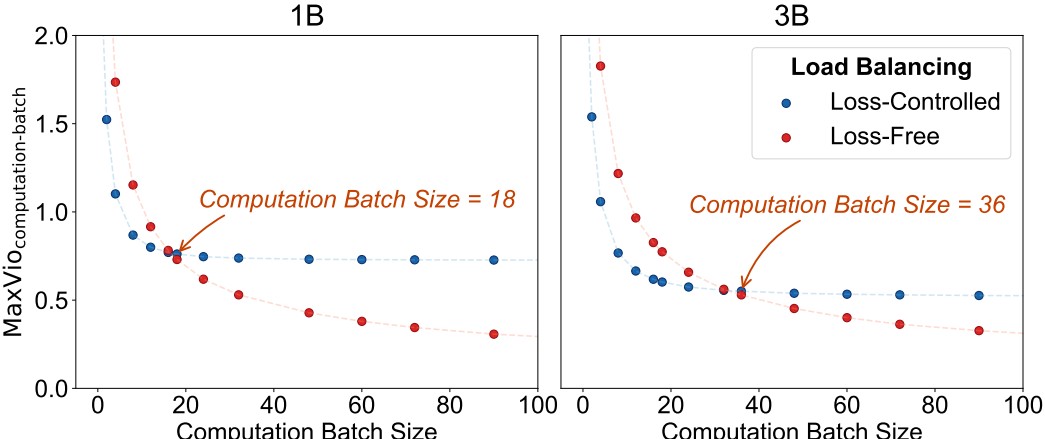

Figure 5: Loss-Free Balancing achieves improved balance compared to auxiliary-loss training as the computation-batch size increases, demonstrating its superiority when a moderately sized computation-batch is utilized.

## 5 DISCUSSION

### 5.1 LOSS-FREE BALANCING IS COMPATIBLE WITH EXPERT PARALLELISM

Extremely large-scale MoE models often employ expert parallelism (Lepikhin et al., 2021) for training or inference, which distributes experts across different devices to reduce memory requirements. In such scenarios, load balance on the data in a single computation step is crucial for efficiency. Due to expert parallelism, each computation step involves `micro_batch_size * ep_data_parallel_size` samples, which we refer to as a **computation batch**. Here, `micro_batch_size` denotes the number of samples processed in one gradient accumulation step on a single device.

Loss-Free Balancing can achieve nearly optimal global load balance, and the load balance in each computation step will get closer to the global load balance as the computation batch size increases. In Figure 5, we examine the computation-batch-level load balance with the MaxVio$_{computation-batch}$ metric. The results show that the load balance of our Loss-Free Balancing always keeps improving as the computation batch size increases, but the load balance of the auxiliary-loss-controlled method approximately maintains a constant level when the computation batch is large. Since expert parallelism will significantly increase the computation batch size by `ep_data_parallel_size` times, Loss-Free Balancing is naturally compatible with large-scale MoE training, and its advantage on the load balance will be further enhanced as the size of expert parallelism increases.

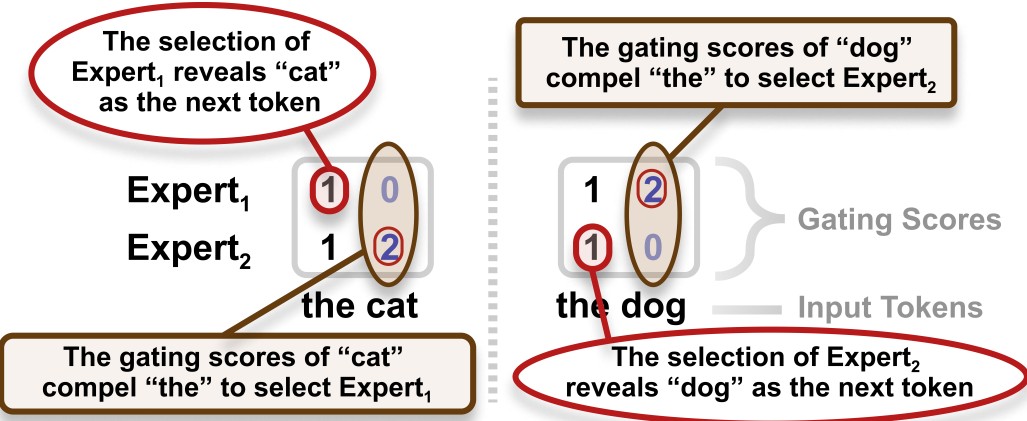

Figure 6: An example of future token leakage in EC. Future tokens can influence the expert assignment of previous tokens. Such an assignment can help previous tokens to infer the identity of their successors.

## 5.2 LOAD BALANCING AND FUTURE TOKEN LEAKAGE

For casual language models, load balancing methods must adhere to the causal constraint of language modeling to avoid future token leakage. While conventional auxiliary-controlled balancing and our Loss-Free Balancing obey this constraint, Expert Choice (EC) (Zhou et al., 2022) violates it. EC ensures perfect load balance by assigning exactly the same number of tokens to each expert. However, this approach inherently leads to a severe issue of future token leakage.

In EC, future tokens can influence the expert assignment of previous tokens. Figure 6 illustrates how information can be easily transmitted within a sequence via such influence. Theoretically, the token assignment of an MoE layer with sparse ratio $R$ (average activated experts per token $K$ divided by total expert number $N$) can leak more than $K \log_2 \frac{1-R}{R}$ bits per token (proof in Appendix D.1). For a 9-layer MoE model with 16 experts and an average of 2 experts per token, this amounts to 50 bits, sufficient for each token to determine its successor's identity.

We designed experiments to demonstrate the existence of future token leakage in realistic model training. **(1)** We reduced the chunk size, within which top-K selection is performed, from 8192 tokens (4 sentences) to 512 (1/4 sentence), with the expectation of exposing such leakage. We observed an abnormal loss drop (about 10%), confirming the presence of leakage. **(2)** We made leakage more difficult by shuffling tokens across chunks in the top-K selection step, and observed that the abnormal loss drop was mitigated. Detailed experimental results on EC's information leakage are provided in Appendix D.2.

Future token leakage is fatal since it destroys the generalization of a model and prevents reliable evaluation of the model performance. Therefore, compared with EC, scaling up an MoE model with our Loss-Free Balancing is safer.

## 6 CONCLUSION

In this work, we introduced **Loss-Free Balancing**, a novel MoE load balance control method that maintains a balanced distribution of expert load without relying on auxiliary losses. By dynamically updating the bias of each expert based on its recent load, Loss-Free Balancing effectively addresses the limitations of traditional auxiliary-loss-based approaches, which can introduce interference gradients during training and potentially impair model performance. Experiments conducted on 1B and 3B MoE models, trained on 100B and 300B tokens respectively, demonstrate that Loss-Free Balancing achieves better model performance and load balance than traditional auxiliary-loss training.

## REPRODUCIBILITY STATEMENT

To ensure the reproducibility of our work, we provide comprehensive details of our model architecture in Appendix A and training hyperparameters, including the learning rate scheduler, initialization, and batch size, in Appendix B. Researchers can easily adapt our method to their own cluster, dataset, and model architecture by following Algorithm 1. We also provide an implementation illustration based on Hugging Face Transformers[2] in the supplementary material.

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

Table 5: Model architecture.

| hyper-parameters | 1B | 3B |
|---|---|---|
| Vocab size | 32064 | 32064 |
| Hidden size | 1024 | 1280 |
| Attention heads | 8 | 10 |
| MoE layers | 9 | 11 |
| Granularity ($\frac{d_\text{ff}}{d_\text{expert}}$) | $\frac{16}{3}$ | 4 |
| Shared experts | 2 | 2 |
| Routed experts | 64 | 64 |
| Activated routed experts | 6 | 6 |

Ashish Vaswani, Noam Shazeer, Niki Parmar, Jakob Uszkoreit, Llion Jones, Aidan N. Gomez, Lukasz Kaiser, and Illia Polosukhin. Attention is all you need. In *NIPS*, pp. 5998–6008, 2017.

Yanqi Zhou, Tao Lei, Hanxiao Liu, Nan Du, Yanping Huang, Vincent Y. Zhao, Andrew M. Dai, Zhifeng Chen, Quoc V. Le, and James Laudon. Mixture-of-experts with expert choice routing. In *NeurIPS*, 2022.

## A  MODEL ARCHITECTURE

We employ the DeepSeekMoE (Dai et al., 2024) architecture as the backbone, which introduces shared experts to mitigate knowledge redundancy among routed experts:

$$\mathbf{h}_t = \mathbf{u}_t + \sum_{i=1}^{N_s} \text{FFN}_i^{(s)}(\mathbf{u}_t) + \sum_{i=1}^{N_r} g_{i,t} \text{FFN}_i^{(r)}(\mathbf{u}_t),\tag{6}$$

where $r$ denotes the routed experts, while $s$ the shared experts. DeepSeekMoE replaces all FFN layers with MoE layers, except the dense FFN layer just after the input embedding layer.

The detailed architecture hyper-parameters are listed in Table 5.

## B  TRAINING SETTINGS

Following the work of Dai et al. (2024), we initialize all learnable parameters with a standard deviation of 0.006, and set the maximum training sequence length to 2048.

For the 1B model, we employ a cosine learning rate scheduler with warmup, setting the learning rate to 1e-3, the minimum learning rate to 1e-4, and the warmup steps to 1000. The training batch size for the 1B model is set to 1152, resulting in a total of 40000 training steps (100B tokens).

For the 3B model, we use a multistep learning rate scheduler with stage steps = [45211, 50862, 56514] and corresponding stage learning rates of [7.8e-4, 2.47e-4, 7.8e-5]. The warmup steps for the 3B model are set to 2000. We use a training batch size of 1728 for the 3B model, resulting in a total of 56514 training steps (200B tokens).

For validation, we leave around 70M tokens from the training corpus as the validation set (30 * 1B_batch_size * max_seq_len = 20 * 3B_batch_size * max_seq_len = 71M tokens).

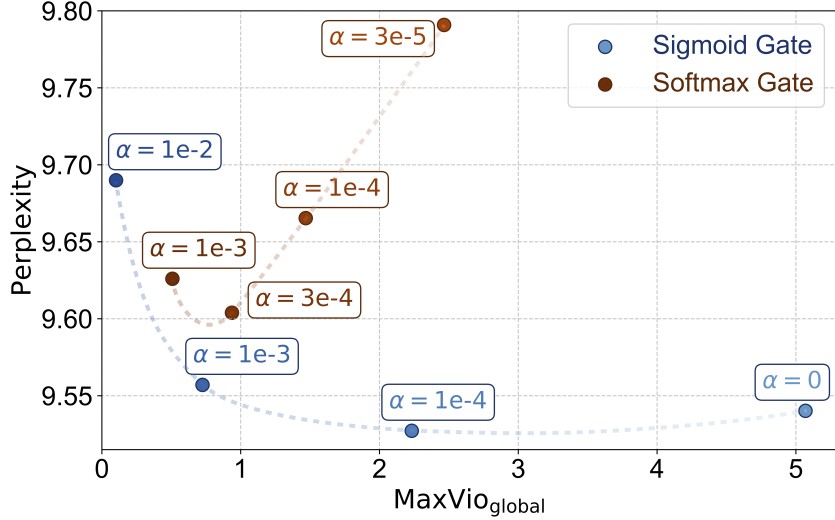

Figure 7: Comparison of the sigmoid gate baseline and the softmax gate baseline. The softmax gate exhibits higher perplexity under similar load balance conditions and is more sensitive to load imbalance compared to the sigmoid gate.

Table 6: For softmax gate, Loss-Free Balancing achieves a slightly lower perplexity while reaching a significantly better load balance compared to the auxiliary-loss training method.

| Load Balancing | Perplexity | MaxVio$_{global}$ |
|---|---|---|
| Loss-Controlled | 9.604 | 0.937 |
| Loss-Free | **9.599** | **0.027** |

## C    EXPERIMENTS WITH SOFTMAX GATE

### C.1    COMPARISON OF SIGMOID GATE BASELINE AND SOFTMAX GATE BASELINE

We compare the sigmoid gate baseline and the softmax gate baseline with varying auxiliary loss coefficients $\alpha$ on a 1B-sized model. As shown in Figure 7, the softmax gate exhibits higher perplexity under similar load balance conditions, and its performance is more sensitive to load imbalance compared to the sigmoid gate.

### C.2    LOSS-FREE LOAD BALANCING WITH SOFTMAX GATE

Adjusting the per-expert bias for the softmax gate is more challenging due to the normalization property of softmax, which makes the score gap between two experts sensitive to the scores of other experts. In such a situation, we choose the $\mathbf{b}_i = \mathbf{b}_i + u * e_i$ variant to maintain load balance, where $u$ is set to 1e-3. For the baseline, we choose $\alpha = 0.0003$, which yields the lowest perplexity for the softmax gate. The results are presented in Table 6, showing that Loss-Free Balancing achieves a slightly lower perplexity while maintaining significantly better load balance compared to the auxiliary-loss training method. Figure 8 confirms that Loss-Free Balancing maintains a superior load balance throughout most of the training process.

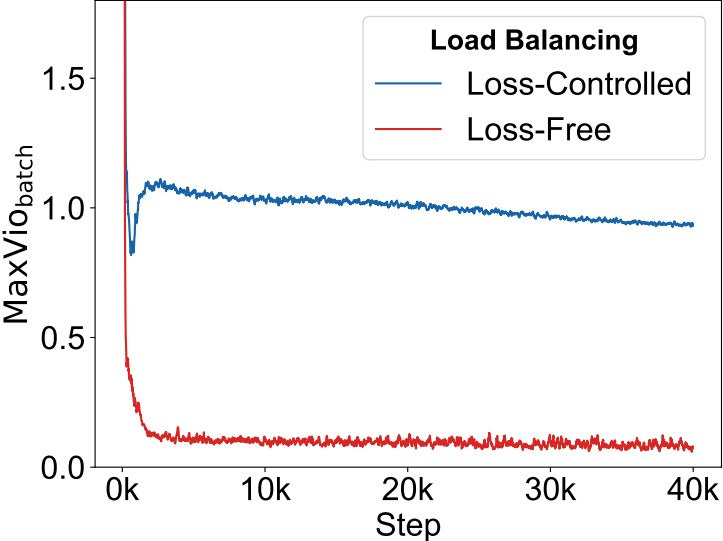

Figure 8: For softmax gate, Loss-Free Balancing maintains a superior load balance throughout most of the training process.

## D FUTURE TOKEN LEAKAGE IN EXPERT CHOICE

### D.1 PROOF FOR THEORETICAL LEAKAGE AMOUNT

Let $R = \frac{K}{N}$ denote the MoE sparsity. Here $K$ denotes the average number of experts activated per token, and $N$ is the total number of experts. For an MoE layer in Expert Choice, the maximum information leakage $I$ (in bits per token), i.e., the information that the combinations of routing allocation can carry is:

$$
\begin{aligned}
I &= \log_2 \left( \frac{\frac{KT}{N}}{T} \right)^N \Big/ T \\
&> N \log_2 \frac{((1 - \frac{K}{N})T)^{\frac{K}{N}T}}{(\frac{K}{N}T)^{\frac{K}{N}T}} \Big/ T \\
&= K \log_2 \frac{1 - R}{R}.
\end{aligned}
\tag{7}
$$

For a model with a sparse ratio $R = \frac{2}{16} = 0.125$ and 9 MoE layers, the total leakage information is more than 50 bits per token.

### D.2 EXPERIMENTAL EVIDENCE

We investigate the potential future token leakage of the Expert Choice by varying the chunk size used for experts' top-$k$ selection, ranging from 512 tokens to 8192 tokens.[3] We train a 2B MoE model on 100B tokens. The results, shown in Table 9, reveal two key findings:

1. Using a small chunk size of 512 leads to an abnormal loss drop, which can be attributed to significant future token leakage. A smaller chunk size allows the model to more easily exploit information from future tokens within the chunk during training.

---

[3]A chunk size of 2048 tokens means performing top-$k$ selection inside a sentence, while 512 tokens correspond to a quarter of a sentence and 8192 tokens to four sentences.

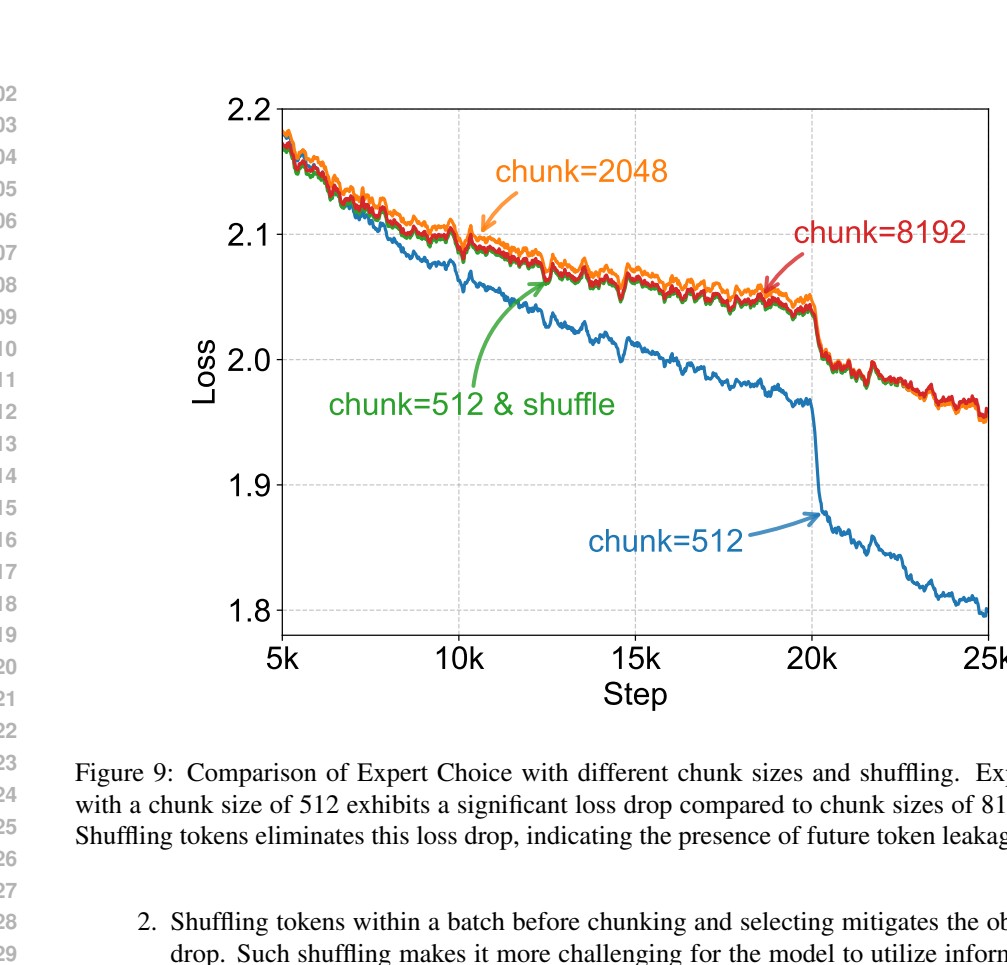

Figure 9: Comparison of Expert Choice with different chunk sizes and shuffling. Expert Choice with a chunk size of 512 exhibits a significant loss drop compared to chunk sizes of 8192 or 2048. Shuffling tokens eliminates this loss drop, indicating the presence of future token leakage.

2. Shuffling tokens within a batch before chunking and selecting mitigates the observed loss drop. Such shuffling makes it more challenging for the model to utilize information leakage, as the future tokens are no longer in their original context. This finding supports the hypothesis that the loss drop originates from the model's accessing and exploiting future token information.

