# OpenReview forum: "Auxiliary-Loss-Free Load Balancing Strategy for Mixture-of-Experts"
_ICLR.cc/2025/Conference — Submitted to ICLR 2025_

### Official Review · Reviewer_QrX9 · 2024-10-23

**Soundness:** 1
**Presentation:** 2
**Contribution:** 1
**Rating:** 3
**Confidence:** 4

**Summary:**

This paper aims to replace the standard auxiliary load-balancing loss in mixture of experts (MoE) models with a new load balancing strategy that does not involve additional losses. This is to ensure tokens are being routed evenly to each expert without introducing gradients that do not directly contribute to achieving the language model objective. The authors also introduce a metric to quantify the degree of load balance in a MoE model.

**Strengths:**

1. There is some originality in viewing the problem of load balancing from the perspective of interfering gradients.

2. The problem and solution proposed are simple and easy to understand.

**Weaknesses:**

1. It is unclear if or how the gradients due to the auxiliary load balancing loss results in the model not achieving its training objective.

2. It is unconvincing that there is a future token leakage when using expert choice. The explanation was brief and relied on Fig. 6 which does not explain how there is a break in the causal constraint of the language modeling. While some empirical evidence is reported, the evidence assumes that "an abnormal loss drop" must be due to a future token leakage.

3. The experimental results are severely lacking. Only one model is used, DeepSeekMoE, instead of the more commonly used Mixtral or traditional Switch transformer. The improvements are marginal even if there is a significant improvement in load balancing (according to their proposed metric).

Minor: in the definition for MaxVio, the $\max_i$ is included in the numerator of fraction which does not make sense as the index i is also involved in $\bar{Load_i}$

**Questions:**

1. Theoretically, can it be shown that an auxiliary loss produces interference gradients that impact model performance?

2. Please clarify further how expert choice leads to future token leakage.

3. Can this method be applied to a more diverse range of MoE models and for computer vision as well. The improvement over the baseline seems marginal. I have doubts about the impact of the method in improving performance.

---

> ### Author Response · Authors · 2024-11-22
>
> 1. **It is unclear if or how the gradients from auxiliary load balancing loss interfere with the model's training objective.**
>
>    We acknowledge that our assertion is primarily based on intuition and requires more rigorous experimental or theoretical validation. We will try to recheck this with more direct experiments and analysis in the next revision.
>
> 2. **The argument about future token leakage in expert choice routing is unconvincing. The explanation lacks detail and Fig. 6 doesn't adequately demonstrate how this breaks causal constraints in language modeling. The empirical evidence assumes token leakage based solely on "abnormal loss drop."**
>
>    + Fig. 6 illustrates a potential information leak mechanism: When processing "cat" in position 2, the model can assign gate scores (0, 2) to make position 1 select expert 1 under Expert Choice. This creates a channel where position 1 can learn about the future token "cat" (e.g., by fixing expert 1 to output hidden states with high "cat" logits, so that when expert 1 selected, the output token must be "cat").
>
>    + In practice, the mechanism may be more complex, potentially encoding future tokens as bits transmitted through multiple Expert Choice MoE layers. And we will emphasize in our revision that this is a potential risk rather than a guaranteed occurrence.
>
>    + Regarding the 0.1 loss drop: While expert choice may offer genuine improvements, such a large gap is suspect. Our concerns align with findings in "Unified Scaling Laws for Routed Language Models" [1], which hypothesizes similar leakage risks.
>
>      > "Our hypothesis is that the re-assignment provides a subtle side channel through which information can be propagated backwards in time, and this can be abused by larger models resulting in the validation loss diverging during training. Adding a shuffling stage ameliorates this issue by introducing a large number of irrelevant elements to the rebalancing process, making it harder to infer behavior of future inputs."
>      >
>      > Appendix B.2.2 in [1]
>
>    + Still, we agree with [1] that "further work is needed to confirm this theory."
>
> 3. **The study's limitation to DeepSeekMoE instead of more widely used models like Mixtral or Switch transformer.**
>
>    Our focus on the DeepSeekMoE-like architecture can be well-justified for several reasons:
>
>    - This design, combining fine-grained routed and shared experts, has demonstrated superior performance over traditional architectures (Mixtral/Switch transformer) in multiple studies: DeepSeekMoE [2], Qwen1.5 MoE [3], and OLMOE paper [4]. (You may find a detailed discussion in [4])
>    - Apart from being successfully adopted by models like Qwen1.5 MoE and XVERSE-MoE-A36B [5], it has also proven highly effective in large-scale implementations such as DeepSeek-V2 (236B) [6] and DeepSeek-Coder-V2 (236B) [7]. Given this, we believe our experiments on this architecture provide sufficient evidence without needing to test old architectures like GShard / Switch Transformers / Mixtral.
>    - Our implementation can be seen as an instantiation of modern MoE architectures, including DeepSeekMoE, QWen and OpenMoE [8], with architecture hyper-parameters optimized for our specific scale.
>    - As mentioned below, we include experiments at larger scale (16B) / with more experts (256).
>
> 4. **The performance gains are modest despite significant load balancing improvements.**
>
>    + Our improvements are statistically significant, exceeding typical random variation (<0.01). Figure 2 demonstrates that when comparing models with aligned $\text{MaxVio}_\text{global}$, we observe a substantial perplexity difference of approximately 0.2.
>
>    + Our 16B-scale MoE model, trained for 1.33T tokens, shows steady improvements across several major benchmarks versus the baseline using auxiliary_loss_alpha=0.001:
>
>      | Benchmark | # shots | Our method | Vanilla Auxiliary Loss |
>      | --------- | ------- | ---------- | ---------------------- |
>      | BBH       | 3-shot  | **39.31**  | 37.31                  |
>      | MMLU      | 5-shot  | **51.77**  | 50.95                  |
>      | C-Eval    | 5-shot  | **52.30**  | 51.56                  |
>      | CMMLU     | 5-shot  | **55.16**  | 54.53                  |
>
>    - Additional experiments with 256 experts at 3B scale (increasing expert count while maintaining other 3B scale hyper-parameters) demonstrate that our method's advantages become more pronounced when the expert number increased:
>
>      | Method                 | Validation Perplexity (**↓**) | ${\textbf{MaxVio}_\textbf{global}}$(**↓**) |
>      | ---------------------- | ----------------------------- | ------------------------------------------ |
>      | Vanilla Auxiliary Loss | 7.38                          | 1.13                                       |
>      | Our Method             | **7.23**                      | **0.10**                                   |

---

> > ### Author Response · Authors · 2024-11-22
> >
> > 5. **Potential application to computer vision.**
> >
> >    Our current focus is language model pretraining. Computer vision applications remain a future research direction.
> >
> > [1] https://arxiv.org/pdf/2202.01169
> >
> > [2] https://arxiv.org/pdf/2401.06066
> >
> > [3] https://qwenlm.github.io/blog/qwen-moe/
> >
> > [4] https://arxiv.org/pdf/2409.02060
> >
> > [5] https://github.com/xverse-ai/XVERSE-MoE-A36B
> >
> > [6] https://arxiv.org/abs/2405.04434
> >
> > [7] https://arxiv.org/abs/2406.11931
> >
> > [8] https://arxiv.org/abs/2402.01739

---

> > > ### Author Response · Authors · 2024-11-22
> > >
> > > Re: Minor issues: Thank you for your kind reminder. We will revise this.

---

> > > > ### Comment · Reviewer_QrX9 · 2024-12-03
> > > >
> > > > Thank you to the authors for their efforts during the rebuttal, your clarifications and additional experimental results are appreciated. However, as the motivation for replacing the standard auxiliary load-balancing loss in MoEs are relatively minor, i.e. how the gradients from auxiliary load balancing loss interfere with the model's training objective and future token leakage in expert choice routing, it is unconvincing that we require a new load balancing strategy. Further, more extensive experimental settings are required to provide sufficient evidence to justify their method. As such, I retain my score.

---

### Official Review · Reviewer_8VSt · 2024-10-30

**Soundness:** 3
**Presentation:** 3
**Contribution:** 2
**Rating:** 5
**Confidence:** 2

**Summary:**

This paper proposes a Logg-free balancing strategy to solve the mixture of expert models' unbalanced expert load problem. The advantage of the approach is that it does not produce any interference gradients.
this paper proved that Loss-Free Balancing achieves better performance and better load balance compared with traditional auxiliary-loss-controlled load balancing strategies

**Strengths:**

This is an interesting research problem and the author aims to develop an efficient solution approach

**Weaknesses:**

1. The motivation and underlying intuition for the proposed approach could be clarified further to enhance understanding.
2. Additional experiments are recommended to demonstrate the robustness of this approach when applied across varying numbers of expert mixtures. A scalable analysis would also be beneficial.
3. The approach would be strengthened with theoretical justification to substantiate its effectiveness.

**Questions:**

1. On Page 4, in Formula 3, the selection and initialization process for b_i remains unclear. It would be helpful to clarify whether b_i  is influenced by K or independent of it. Additionally, in the adjustment process—where each bias b_i is iteratively modified based on the load of the corresponding expert—it is unclear by what amount b_i  should be increased or decreased and according to which theoretical basis this adjustment is made.
2. On Page 8, it is stated that 'the load balance of our Loss-Free Balancing consistently improves as the computation batch size increases.' Is there an upper bound to this improvement?
3. Is there a specific reason for selecting 9 MoE layers for testing?

---

> ### Author Response · Authors · 2024-11-22
>
> 1. **The motivation and underlying intuition for the proposed approach could be clarified further to enhance understanding.**
>
>    We acknowledge this feedback and will enhance the explanation of Loss-Free Balancing's motivation and mechanics, particularly focusing on how it achieves an improved trade-off between load balance and performance.
>
> 2. **Additional experiments are recommended to demonstrate the robustness of this approach when applied across varying numbers of expert mixtures. A scalable analysis would also be beneficial.**
>
>    We appreciate this suggestion and have conducted two additional sets of experiments:
>
>    + At 3B scale, we increase the expert number to 256 (other hyperparameters unchanged), Our Loss-Free Balancing demonstrated enhanced advantage:
>
>      |                        | Validation Perplexity(**↓**) | ${\textbf{MaxVio}_\textbf{global}}$(**↓**) |
>      | ---------------------- | ---------------------------- | ------------------------------------------ |
>      | Vanilla Auxiliary Loss | 7.38                         | 1.13                                       |
>      | Our Method             | **7.23**                     | **0.10**                                   |
>
>    + At 16B scale and 1.33T training tokens, we compare our method against the vanilla auxiliary loss baseline (auxiliary_loss_alpha=0.001), demonstrating steady improvements across multiple benchmarks:
>
>      | Benchmark (Acc.) | # shots | Our Method | Vanilla Auxiliary Loss |
>      | ---------------- | ------- | ---------- | ---------------------- |
>      | BBH              | 3-shot  | **39.31**  | 37.31                  |
>      | MMLU             | 5-shot  | **51.77**  | 50.95                  |
>      | C-Eval           | 5-shot  | **52.30**  | 51.56                  |
>      | CMMLU            | 5-shot  | **55.16**  | 54.53                  |
>
> 3. **The selection and initialization process in Formula 3.**
>
>    The process is currently stated in Algorithm 1 (page 4) with supporting explanations in the surrounding text. We will revise this section for greater clarity.
>
> 4. **Regarding the stated improvement in load balance with increased computation batch size (Page 8): Is there an upper bound?**
>
>    As illustrated in Figure 5, our method and vanilla auxiliary loss exhibit different convergence behavior:
>
>    + For our method, the MaxVio asymptotically approaches 0 as the batch size tends toward infinity. (the $\text{MaxVio}_\text{global}$ equals to a computation batch size = about 30k based on our testing protocol, for which our method is below 0.1)
>    + For vanilla auxiliary loss, MaxVio may not come close to 0 when the computation batch increases. (the $\text{MaxVio}_\text{global}$ equals to a computation batch size = about 30k based on our testing protocol, for which the vanilla auxiliary loss still has high MaxVio.)
>
> 5. **Is there a specific reason for selecting 9 MoE layers for testing?**
>    This hyperparameter was selected for our 1B baseline model and inherited for consistency in evaluating Loss-Free Balancing. (And for the 3B model, we use 11 layers)
>
> 6. **Is any theoretical proof available for this MoE method?**
>    The inherent complexity of deep neural networks makes theoretical proofs challenging, not only for our approach but also for traditional vanilla auxiliary loss methods. In this paper, we prioritize experimental validation and have observed convincing improvements, as outlined both above and throughout our work.

---

### Official Review · Reviewer_TZLQ · 2024-11-03

**Soundness:** 2
**Presentation:** 1
**Contribution:** 2
**Rating:** 5
**Confidence:** 4

**Summary:**

The paper introduces Loss-Free Balancing, a new load balancing strategy for MoEs that avoids auxiliary loss, which traditionally introduces interference gradients and hampers model performance. The motivation stems from the need to balance expert loads in MoE models to prevent routing collapse and computational overhead, issues that current auxiliary-loss methods attempt to address but with performance trade-offs. The major research question is whether a balancing strategy can maintain expert load distribution without harming model performance. The proposed method adjusts each expert's routing score using a dynamically updated bias based on recent loads, promoting balanced expert utilization without adding interference gradients. Experiments on 1B and 3B parameter MoE models trained on abundant datasets show that Loss-Free Balancing achieves better load balance and improved validation perplexity compared to traditional methods, making it a promising approach for scaling large language models while preserving efficiency and performance.

**Strengths:**

This paper has the following strengths:

1. The proposed Loss-Free Balancing method eliminates the need for auxiliary loss, which traditionally adds undesirable interference gradients. This results in a cleaner training signal focused solely on the primary language modeling objective, potentially enhancing overall model performance.

2. By dynamically adjusting biases for each expert based on recent load data, the method ensures a balanced expert load without compromising model efficiency.

3. The strategy is compatible with expert parallelism, a key feature for training extremely large MoE models across multiple devices. This makes it highly suitable for scaling up model sizes while maintaining efficient load balancing.

4. Unlike the Expert Choice (EC) method, which risks future token leakage, Loss-Free Balancing maintains causal constraints in language modeling, thus preserving the integrity of model generalization.

**Weaknesses:**

This paper has the following weaknesses:

1. I am at first astonished by the short reference list of this paper, as the authors only cited 10 papers. Clearly, this paper did a very bad job on surveying the related work, including the various auxilliary-loss-based balancing methods, the major improvement of MoEs, the current MoE-based LLMs. Normally, I would list a few of the works for your reference, but the authors missed too many so I do not know where to start. I would strongly suggest the authors to check other MoE-LLM papers to improve the related work part.

2. The authors seemed to enlarge the figures in this paper in order to make the paper length reach 9 pages, which result in a disproportionate paper layout, and the figures look abrupt.

3. The architecture design of the MoE-based LLMs are versatile. This paper only demonstrates its effectiveness on a small DeepSeek MoE model, while leaving other MoE models unvisited, such as Mistral, QWen, LLaMA-MoE, OpenMoE.

4. The improvement in performance seem trivial, which challenges the motivation of this study: why do we need a loss-free balancing at all.

**Questions:**

Please see my questions in Weakness column.

---

> ### Author Response · Authors · 2024-11-22
>
> 1. **Reference Scope**
>
>    We acknowledge that our focused discussion on MoE load balancing may have led to a limited reference scope. We will expand our literature review to include broader MoE-related works in the revised version. We welcome specific reference suggestions to enhance the comprehensiveness of our discussion.
>
> 2. **Figure Adjustments**
>
>    We will optimize figure sizes in the revised manuscript.
>
> 3. **Architectural Comparisons**
>
>    Our focus on the DeepSeekMoE-like architecture is well-justified for several reasons:
>
>    - This architecture, featuring fine-grained routed experts and shared experts, proved to be superior by DeepSeekMoE [1], Qwen1.5 MoE [2], and OLMOE paper [3]. (You can find a detailed discussion in [3]). Apart from being successfully adopted by models like Qwen1.5 MoE and XVERSE-MoE-A36B [4], it has also proven highly effective in large-scale implementations such as DeepSeek-V2 (236B) [5] and DeepSeek-Coder-V2 (236B) [6]. Given this, we believe our experiments on this architecture provide sufficient evidence without needing to test old architectures like GShard [7] / Switch Transformers [8] / Mixtral.
>    - Our implementation can be seen as an instantiation of modern MoE architectures, including DeepSeekMoE, QWen and OpenMoE, with architecture hyper-parameters optimized for our specific scale.
>    - Regarding LLaMA-MoE: Recent studies (OLMOE [3], Skywork-MoE [9]) demonstrate that MoE models trained from scratch will ultimately outperform upcycled models after a decent number of training tokens, making it less relevant for our pretraining-focused study.
>
> 4. **Performance Improvements**
>
>    The significance of our improvements is demonstrated through multiple lines of evidence:
>
>    - The perplexity improvements exceed random variation (typically <0.01), indicating statistical significance. Figure 2 also shows a perplexity difference of about 0.2 when aligned with $\text{MaxVio}_\text{global}$.
>
>    - Our 16B-scale MoE model, trained for 1.33T tokens, shows steady improvements across several major benchmarks versus the baseline using auxiliary_loss_alpha=0.001:
>
>      | Benchmark | # shots | Our method | Vanilla Auxiliary Loss |
>      | --------- | ------- | ---------- | ---------------------- |
>      | BBH       | 3-shot  | **39.31**  | 37.31                  |
>      | MMLU      | 5-shot  | **51.77**  | 50.95                  |
>      | C-Eval    | 5-shot  | **52.30**  | 51.56                  |
>      | CMMLU     | 5-shot  | **55.16**  | 54.53                  |
>
>    - Additional experiments with 256 experts at 3B scale (increasing expert count while maintaining other 3B scale hyper-parameters) demonstrate that our method's advantages become more pronounced when expert number increased:
>
>      | Method                 | Validation Perplexity (**↓**) | ${\textbf{MaxVio}_\textbf{global}}$(**↓**) |
>      | ---------------------- | ----------------------------- | ------------------------------------------ |
>      | Vanilla Auxiliary Loss | 7.38                          | 1.13                                       |
>      | Our Method             | **7.23**                      | **0.10**                                   |
>
> [1] https://arxiv.org/pdf/2401.06066
>
> [2] https://qwenlm.github.io/blog/qwen-moe/
>
> [3] https://arxiv.org/pdf/2409.02060
>
> [4] https://github.com/xverse-ai/XVERSE-MoE-A36B
>
> [5] https://arxiv.org/abs/2405.04434
>
> [6] https://arxiv.org/pdf/2406.11931
>
> [7] https://arxiv.org/abs/2006.16668
>
> [8] https://arxiv.org/abs/2101.03961
>
> [9] https://arxiv.org/pdf/2406.06563

---

### Official Review · Reviewer_tLSf · 2024-11-05

**Soundness:** 2
**Presentation:** 2
**Contribution:** 2
**Rating:** 3
**Confidence:** 4

**Summary:**

The authors propose an alternative loss-free method for load balancing of experts during MoE training. Load imbalance is a critical issue in MoE training as it can lead to expert collapse or increased utilization of some experts over the others. The proposed method achieves load balancing by dynamically applying expert-wise biases on routing scores according to their recent load, avoiding interfering gradients. The added bias is designed to only affect the top-k selection without changing the routing weights for combining the selected experts.

The loss-free load balancing approach is applied to DeepSeekMoE models with sizes 1B and 3B. The authors report perplexity on the validation set vs Maximal Violation scores.

**Strengths:**

- The paper is well-written and the method is clearly explained
- Good visualizations
- Simple approach that should be very easy to test and cheaper to compute than the conventionally used load-balancing loss

**Weaknesses:**

- I am concerned over the validity of the claims. The empirical evaluations are very limited constrained to two DeepSeekMoE models and perplexity differences among the models and the baselines are at the level of 0.05 difference. Is this difference in perplexity significant?

- The evaluation is limited to language modelling and perplexity values. It would be better to see the actual effect of the loss-free load balancing on other downstream tasks such as MMLU or GLUE.

- The proposed Max Violation (MaxVio) score subtracts the mean load from the maximum load an expert receives, which highlights the worst-case scenario of load imbalance. Since MaxVio focuses on the maximum load, it can be highly sensitive to outliers. A single batch with unusually high token routing to one expert can disproportionately affect the MaxVio calculation, making it appear as though there is a significant imbalance even if most batches are well-balanced.

- To make the comparison fair, it would be useful to also see the load balancing (loss) values (without back propagating) and see how the loss-free variant behaves on that score as the training continues (e.g., a plot of load balancing loss over training steps for both methods).

**Questions:**

The load-balancing loss brings the advantage of load balancing to the inference stage by directly affecting the routing weights? How does the proposed method behave during inference.

---

> ### Author Response · Authors · 2024-11-22
>
> 1. **Is the improvement of Loss-Free Balancing significant? Could more benchmarks be reported?**
>
>    + The improvement is statistically significant, as the observed perplexity differences exceed typical random variation (which is usually below 0.01). Furthermore, Figure 2 demonstrates that when comparing models with aligned $\text{MaxVio}_\text{global}$, we observe a substantial perplexity difference of approximately 0.2.
>
>    + We conducted evaluation on a 16B-scale MoE model trained with our method versus a baseline using auxiliary_loss_alpha=0.001. Both models were trained for 1.33T tokens, and our method shows steady improvement over the baseline (using auxiliary_loss_alpha=0.001) across multiple benchmarks:
>
>      | Benchmark (Acc.) | # shots | Our method | Vanilla Auxiliary Loss |
>      | ---------------- | ------- | ---------- | ---------------------- |
>      | BBH              | 3-shot  | **39.31**  | 37.31                  |
>      | MMLU             | 5-shot  | **51.77**  | 50.95                  |
>      | C-Eval           | 5-shot  | **52.30**  | 51.56                  |
>      | CMMLU            | 5-shot  | **55.16**  | 54.53                  |
>
> 2. **Can more models be reported?**
>
>    + Here, by the word 'DeepSeekMoE architecture', we refer to a mainstream MoE architecture design incorporating (a) fine-grained routed experts and (b) shared experts. This has been proved effective by 236B DeepSeek-V2 [1] and DeepSeek-Coder-V2 [2]. This architecture has also been adopted by current models including Qwen1.5 MoE [3] and XVERSE-MoE-A36B [4], and its benefits are thoroughly analyzed in the OLMOE paper [5] (you may refer to this for a detailed discussion). Given this, we believe our experiments on this architecture provide sufficient evidence without needing to test old architectures like GShard [6] / Switch Transformers [7].
>
>    + Beyond the 16B-scale results presented above, we've conducted additional experiments with the expert number increased to 256 at 3B scale (increasing the expert number while keeping other hyperparameters the same as 3B setting), where Loss-Free Balancing enlarges its advantage on perplexity:
>
>      |                        | Validation Perplexity(**↓**) | ${\textbf{MaxVio}_\textbf{global}}$(**↓**) |
>      | ---------------------- | ---------------------------- | ------------------------------------------ |
>      | Vanilla Auxiliary Loss | 7.38                         | 1.13                                       |
>      | Our Method             | **7.23**                     | **0.10**                                   |
>
> 3. **Is Max Violation a proper metric for MoE's load balance? Should auxiliary loss be considered as a load balance metric for Loss-Free Balancing?**
>
>    + Load balancing in MoE is primarily an engineering consideration rather than a mathematical abstraction. It's crucial for expert parallelization, where processing speed is bottlenecked by the most loaded device. Therefore, Max Violation directly measures what matters most in practical implementations.
>    + Max Violation is more appropriate than auxiliary loss for measuring load imbalance because it directly reflects processing speed. Auxiliary loss, incorporating routing scores, doesn't purely measure load balance and lacks clear meaning (for example, two models with identical routing patterns may have different auxiliary loss values due to different routing scores).
>
> 4. **For the "Questions" section:**
>    As 3. mentioned, the Max Violation metric effectively captures the load balance characteristics relevant to practical implementations. In the paper, we have reported our experimental results on the validation set (Section 4.2 & Section 5.1), which show the load balance situation of auxiliary-loss method and our method.
>
> [1] https://arxiv.org/abs/2405.04434
>
> [2] https://arxiv.org/abs/2406.11931
>
> [3] https://qwenlm.github.io/blog/qwen-moe/
>
> [4] https://github.com/xverse-ai/XVERSE-MoE-A36B
>
> [5] https://arxiv.org/pdf/2409.02060
>
> [6] https://arxiv.org/abs/2006.16668
>
> [7] https://arxiv.org/abs/2101.03961

---

> > ### Comment · Reviewer_TZLQ · 2024-11-27
> > **Thanks for the response**
> >
> > Dear authors,
> >
> > Thanks for the responses and I am ok with most of the replies. Therefore, I would increase my score from 3 to 5.
> >
> > Best,

---

> > ### Comment · Reviewer_tLSf · 2024-11-29
> >
> > I would like to thank the authors for their response which addressed some of my concerns, however, most of my concerns remain unresolved and the paper requires significant improvement in experimental evaluation to be considered for ICLR.

---

### Meta-Review · Area_Chair_p4x6 · 2024-12-18

**Metareview:**

After reviewing the authors' rebuttal and considering the discussion phase, all reviewers maintained reservations about this submission, resulting in final ratings of 3 (Reviewer tLSf), 5 (Reviewer TZLQ), 5 (Reviewer 8VSt), and 3 (Reviewer QrX9).

While the rebuttal partially addressed some concerns, notably leading Reviewer TZLQ to raise their score to 5, the reviewers collectively concluded that the remaining weaknesses outweigh the strengths of the work. Important issues, as identified during the review process, limit the overall contribution and impact of the paper. Regrettably, given the current state of the submission and the lack of a compelling consensus in favor of its acceptance, I recommend rejection.

**Additional Comments On Reviewer Discussion:**

All the reviewers felt that the weaknesses still outweigh the strengths of the work post rebuttal.

---

### Decision · Program_Chairs · 2025-01-22

Reject